# Learning and analyzing vector encoding of symbolic representations

**Roland Fernandez, Aslı Çelikyılmaz**
Microsoft Research AI
Redmond, WA 98052, USA
{rfernand,aslicel}@microsoft.com

**Paul Smolensky**
Microsoft Research AI & Johns Hopkins University
Redmond, WA 98052, USA & Baltimore, MD 21218, USA
psmo@microsoft.com, smolensky@jhu.edu

**Rishabh Singh**
Google Brain
Mountain View, CA, USA
rishabh.iit@gmail.com

## Abstract

We present a formal language with expressions denoting general symbol structures and queries which access information in those structures. A sequence-to-sequence network processing this language learns to encode symbol structures and query them. The learned representation (approximately) shares a simple linearity property with theoretical techniques for performing this task.

## 1 Overview: S-Lang, S-Net and S-Rep

Embedding words and sentences in vector spaces has brought many symbolic tasks (especially in NLP) within the scope of deep neural network (DNN) models (Hinton, 1988; Palangi et al., 2016; Pollack, 1990; Socher et al., 2010; Weston et al., 2015). In general, DNNs may be expected to benefit if they can incorporate some of the power of symbolic computation without compromising the power of deep learning. The problem of embedding general symbol structures in vector spaces, and performing symbolic computation with these vectors, has been addressed theoretically, but these methods can require very large embedding spaces — e.g., Tensor Product Representation, TPR (Lee et al., 2016; Smolensky, 1990; Smolensky & Legendre, 2006) — or major error-correction/clean-up processes — e.g., Holographic Reduced Representation, HRR (Crawford et al., 2016; Plate, 1993; 2002; 2003) (Sec. 4; see also Kanerva (2009); Touretzky (1990)). We show here (Sec. 3) that deep learning can itself discover satisfactory methods of embedding general symbol structures, methods that operate in relatively small vector spaces without need for clean-up processing.

We define a general formal language scheme in which expressions denote symbol structures. Such a language will be called an S-Lang (Sec. 2). Information within these structures is accessed by evaluating query expressions within the language. The model that learns to encode structure-denoting expressions and to evaluate queries over these structures (Zaremba & Sutskever, 2014) is a simple bidirectional encoder-decoder model that operates on symbols in the formal language one at a time (Cho et al., 2014). We call such a model an S-Net (Sec. 3), and call the vector embedding of an S-Lang learned by an S-Net an S-Rep.

Our S-Net learns to evaluate S-Lang expressions with a high degree of accuracy (Sec. 3). Furthermore, upon analysis, the S-Rep that S-Net learns turns out to exhibit a simple linearity property — the Superposition Principle — that is crucial to both theoretical models, TPR and HRR (Sec. 4).

## 2 The task: Embedding general symbolic structures in vector spaces and accessing their contents

*Symbol-structure-denoting expressions: Structural-role binding*. In general, a symbol structure can be characterized as a set of symbols each bound to a role that it plays in the structure (Newell,

Table 1: Example input/output pairs of the S-NET model

| Expression | Value | Type |
|---|---|---|
| `as:U` | `as:U` | binding |
| `qf:N ? N` | `qf` | unbind |
| `qf:N ? X` | `$` | unbind (not found) |
| `(ao:N & ax:F & wh:A ? F):K` | `ax:K` | 3-bind, unbind, rebind |
| `((((sf:W & fr:V):N):R):R):Y ? Y ? R ? R ? N` | `sf:W & fr:V` | 4-nested, unbind |

Table 2: Performance and hyperparameters of the trained S-NET model

| Accu-racy | Test per-plexity | Train loss | Mini-batches | Batch size | Hidden dimension | Drop-out | Learn-ing rate | Optim-izer | Attention; Beam |
|---|---|---|---|---|---|---|---|---|---|
| 96.16 | 1.02 | 0.187 | 54K | 128 | 128 | 0.2 | 0.001 | Adam | None |

1980, 141). The method is applicable to any type of symbol structure, but we focus on binary trees here. The simple binary tree $\mathcal{T} = [\text{a } [\text{b c}]]$ consists of the symbols a, b, c respectively bound to the roles $R_0, R_{01}, R_{11}$, where $R_{01}$ is the role of left-child ('0') of right-child ('1') of root, etc. Symbolic structural roles are typically recursive. The recursive character of binary tree roles can be seen by viewing $R_{01}$ as the symbol $R_0$ bound to the role $R_1$. In the simple formal language we develop here, S-LANG, the tree $\mathcal{T}$ will be denoted by the expression `a:L & b:L:R & c:R:R`, where L, R respectively abbreviate the roles $R_0, R_1$. The grammar of S-LANG is shown in Fig. 1.

```
<exp>        ::= <nil> | <filler-id> | <binding> | <unbinding> | <superimpose> | ( <exp> )
<binding>    ::= <exp> : <role-id>
<unbinding>  ::= <exp> ? <role-id> [ ? <role-id> ...]
<superimpose> ::= <exp> & <exp>
<filler-id>  ::= <id that starts with lowercase letter>
<role-id>    ::= <id that starts with uppercase letter>
<nil>        ::= $
```

Figure 1: Grammar of S-LANG: expressions denoting symbol structures and access queries.

*Query-denoting expressions: Structural-role unbinding.* A minimal requirement for a vector embedding a symbol structure is that it be possible to extract, with vector computation, the (embedding of) the symbol that is bound to any specified role. In S-LANG (see Fig. 1), the query denoted ? S asks for the structure (possibly a single symbol), bound to the role denoted S; this is role-unbinding. Thus the expression `a:L & b:L:R & c:R:R ? L` asks for the structure filling the role L in $\mathcal{T}$, i.e., the structure forming the left child of the root, which in $\mathcal{T}$ happens to be a single symbol, a. Similarly, the expression `a:L & b:L:R & c:R:R ? L:R` asks for the left child of the right child of the root, and so has the value b. Afn S-LANG query can return a structure rather than a single symbol. The expression `a:L & b:L:R & c:R:R ? R` asks for the right child of the root of $\mathcal{T}$, which is the structure `b:L & c:R`, denoting $\mathcal{T}$'s right sub-tree, $[\text{b c}]$.

*Expressions combining querying and structure-building.* The general expression in S-LANG allows structure that is returned by queries to be used to build new structures. Table 1 provides examples of expressions correctly evaluated by S-NET.

## 3    THE S-NET MODEL AND EXPERIMENTAL RESULTS

S-NET is a standard bidirectional encoder-decoder network where the output of the bidirectional LSTM encoder is the S-REP embedding of the input S-LANG expression. The S-REP vector is then fed as input to an LSTM decoder. Some implementation details are given in Table 2, which also gives the results of training S-NET on randomly-selected input/output pairs.

## 4    ANALYSIS OF S-REP: THE SUPERPOSITION PRINCIPLE

*The Superposition Principle in theoretical structure-embedding schemes.* Theoretical solutions to performing the task defined in Sec. 2 were proposed in the previous generation of neural network

modeling. Two general solutions, TPR and HRR, were introduced in Sec. 1. The TPR embedding of a symbol structure $\mathcal{S}$ with symbols $\{\mathbf{s_k}\}$ respectively bound to roles $\{\mathbf{r_k}\}$ is $\mathrm{TPR}(\mathcal{S}) \equiv \sum_k \mathbf{s_k} \otimes \mathbf{r_k}$, where $\otimes$ denotes the tensor (generalized outer) product and $\{\mathbf{s_k}\}$ and $\{\mathbf{r_k}\}$ are embeddings of the symbols and roles, with respective dimensions $\sigma$ and $\rho$; the dimension of the TPR itself is then $\sigma\rho$.

If the role-embedding vectors $\{\mathbf{r_k}\}$ are linearly independent, when collected together they form an invertible matrix $\mathbf{R}$; the rows of $\mathbf{R}^{-1}$ are the "unbinding" vectors $\{\mathbf{u_k}\}$: $\mathbf{r_k} \cdot \mathbf{u_j} = \delta_{kj}$ so these vectors can be used to unbind the roles in a TPR. The symbol that fills role $\mathbf{r_k}$ in structure $\mathcal{S}$ is exactly the symbol $\mathbf{s_k}$ with embedding $\mathbf{s_k} = \mathrm{TPR}(\mathcal{S}) \cdot \mathbf{u_k}$.

A crucial property of TPR is that the embedding of a structure is the *sum* over embeddings of its symbols. This is TPR's Superposition Principle. This is what enables extraction of symbols from any binding: $(\sum_k \mathbf{s_k} \otimes \mathbf{r_k}) \cdot \mathbf{u_j} = \sum_k \mathbf{s_k} \otimes (\mathbf{r_k} \cdot \mathbf{u_j}) = \mathbf{s_j}$ since $\mathbf{r_k} \cdot \mathbf{u_j} = \delta_{kj}$.

HRRs are essentially contracted TPRs (Smolensky & Legendre, 2006, 260). The equation defining TPR($\mathcal{S}$) also defines HRR($\mathcal{S}$), provided $\otimes$ is reinterpreted to denote circular convolution: $[\mathbf{a} \otimes \mathbf{b}]_\mu = \sum_\nu [\mathbf{a}]_\nu [\mathbf{b}]_{\mu-\nu}$. Assuming the elements of the $\{\mathbf{r_k}\}$ are randomly (typically, normally) distributed, each role-embedding vector $\mathbf{r_k}$ can be used as its own unbinding vector. However the HRR unbinding equation holds only approximately: $\mathbf{s_k} = \mathrm{HRR}(\mathcal{S}) \cdot \mathbf{u_k} + \mathrm{noise}$. This noise must be eliminated by 'clean-up' processes. Note that, like TPR, HRR obeys the Superposition Principle.

*Testing the Superposition Principle in the learned representation.* As a test of whether the Superposition Principle holds of S-REP, let $\mathbf{v}(\texttt{expr})$ denote the S-REP vector embedding of S-LANG expression expr), and consider expressions containing two symbol/role bindings, such as aa:A & bb:B. Then if the Superposition Principle holds, we have[1]:

1. $[\mathbf{v}(\texttt{aa:A \& bb:B}) - \mathbf{v}(\texttt{aa:A \& cc:C})] - [\mathbf{v}(\texttt{dd:D \& bb:B}) - \mathbf{v}(\texttt{dd:D \& cc:C})] = \mathbf{0}$.

2. $[\mathbf{v}(\texttt{aa:A \& bb:B}) - \mathbf{v}(\texttt{aa:A \& cc:C})] - [\mathbf{v}(\texttt{xx:X \& uu:U}) - \mathbf{v}(\texttt{xx:X \& vv:V})] \neq \mathbf{0}$.

We examine the Euclidean length of vectors of the form given on the LHS of Eq. 1, which we expect to not be exactly 0, but small — small, for example, relative to the LHS of Eq. 2. Fig. 2 shows that this is true: the AUC = 1.0 to within less than $10^{-16}$.

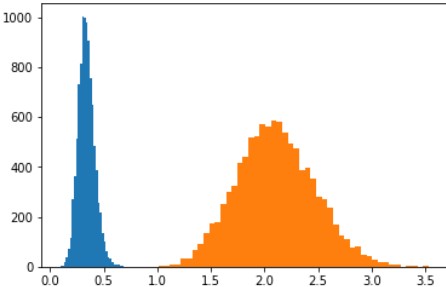

Figure 2: Distribution of lengths of vectors of the form of the LHS of Eqs. 1 and 2.

## 5   CONCLUSION

A standard bidirectional encoder-decoder model can generate vector embeddings of expressions denoting complex symbol structures and can successfully query the content of such representations. Like theoretical techniques for accomplishing this, the learned representation obeys the Superposition Principle (approximately; at least within the manifold of embeddings of two-binding expressions).

---

[1]The simpler equation $\mathbf{v}(\texttt{aa:A \& bb:B}) - [\mathbf{v}(\texttt{aa:A}) + \mathbf{bb:B}] = \mathbf{0}$ does not hold in S-REP; it appears that the manifolds of one- and two-binding embeddings are distinct. Eq. 1 stays within the latter. Eq. 1 is analogous to the famous equation $[\mathbf{v}(\texttt{king}) - \mathbf{v})(\texttt{man})] - [\mathbf{v}(\texttt{queen}) - \mathbf{v}(\texttt{woman})] \approx 0$ (Mikolov et al., 2013).

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
