# OpenReview forum: "LEARNING AND ANALYZING VECTOR ENCODING OF SYMBOLIC REPRESENTATION"
_ICLR.cc/2018/Workshop — Accept_

### Official Review · AnonReviewer2 · 2018-03-09
**Back to symbols**

**Rating:** 8
**Confidence:** 4

**Review:**

The paper proposes a method to encode and decode symbolic representations in vectors using neural networks. This is a very important problem in neural networks. Yet, the paper forgets to mention that this topic has been revitalized some years ago within the distributed tree kernels and within the compositional distributional semantics model (Zanzotto, Ferroni, Baroni, 2015).

---

### Official Review · AnonReviewer3 · 2018-03-11
**not sure I understand this paper**

**Rating:** 6
**Confidence:** 1

**Review:**

I am not sure  I understand this paper.

---

### Official Review · AnonReviewer1 · 2018-03-16
**Good paper. Accept.**

**Rating:** 7
**Confidence:** 4

**Review:**

Summary:

This paper studies the power of generic sequence to sequence models in representing general symbol structures. To do that, authors propose a simple formal language and generate structures and queries in this formal language. Then they train a sequence to sequence model with bidirectional LSTM encoder and an LSTM decoder. This simple model achieves high accuracy and also demonstrates superposition principle.


My Comments:

This is an interesting study which is worth reporting in the workshop track.

1. What is the effect of length of the expression? Does the performance degrade after certain length?
2. In Query-denoting expressions paragraph, correct “Afn S-LANG ..”

---

### Decision · Program_Chairs · 2018-03-20
**ICLR 2018 Workshop Acceptance Decision**

**Decision:**

Accept

**Comment:**

Congratulations, your paper was accepted to the ICLR workshop.